# From interaction networks to interfaces, scanning intrinsically disordered regions using AlphaFold2

Hélène Bret[1], Jinmei Gao[1], Diego Javier Zea[1], Jessica Andreani [1] & Raphaël Guerois [1]

The revolution brought about by AlphaFold2 opens promising perspectives to unravel the complexity of protein-protein interaction networks. The analysis of interaction networks obtained from proteomics experiments does not systematically provide the delimitations of the interaction regions. This is of particular concern in the case of interactions mediated by intrinsically disordered regions, in which the interaction site is generally small. Using a dataset of protein-peptide complexes involving intrinsically disordered regions that are non-redundant with the structures used in AlphaFold2 training, we show that when using the full sequences of the proteins, AlphaFold2-Multimer only achieves 40% success rate in identifying the correct site and structure of the interface. By delineating the interaction region into fragments of decreasing size and combining different strategies for integrating evolutionary information, we manage to raise this success rate up to 90%. We obtain similar success rates using a much larger dataset of protein complexes taken from the ELM database. Beyond the correct identification of the interaction site, our study also explores specificity issues. We show the advantages and limitations of using the AlphaFold2 confidence score to discriminate between alternative binding partners, a task that can be particularly challenging in the case of small interaction motifs.

Protein interactions are crucial for a vast number of processes in living organisms. Strong evidence points to the biological importance of interactions mediated by intrinsically disordered protein regions (IDRs), such as short linear motifs, in particular for regulation, transport and signaling, and in a number of human pathologies[1–3]. Established resources exist to identify already annotated binding motifs, such as the Eukaryotic Linear Motif (ELM) repository[4], to visualize evolutionary properties[5] and to screen full protein sequences for disordered stretches that might fold upon binding, as with the IUPred server[6].

Protein interactions are connected within complex networks called interactomes, which can be derived from large amounts of experimental data such as proteomics. Much effort has been invested into mapping and modeling interactions at the scale of these interactomes[7,8]. In these networks, most protein-protein interactions evolve under negative selection to maintain function and many of them can rewire[9], although at different evolutionary rates: stable protein complexes evolve more slowly than most domain-motif interactions[10]. Interactions in evolutionarily old, housekeeping protein complexes are conserved across different contexts (cell types, tissues and conditions) while evolutionarily young interactions and those mediated by disordered regions are more versatile[11,12]. Evolutionary conservation has long been recognized as relevant to detect binding motifs in disordered regions, as reviewed in[13]; however, the

[1]Université Paris-Saclay, CEA, CNRS, Institute for Integrative Biology of the Cell (I2BC), 91198 Gif-sur-Yvette, France. ✉e-mail: jessica.andreani@cea.fr; raphael.guerois@cea.fr

quality of the multiple sequence alignment (MSA) used for detection is particularly crucial[14].

AlphaFold2 revolutionized structure prediction for single proteins by leveraging deep learning approaches to extract signal from MSAs and output protein atomic 3D coordinates in an end-to-end manner[15]. AlphaFold2 structure predictions for the entire human proteome[16] hinted that low prediction quality could pinpoint regions likely to be intrinsically disordered. Subsequent studies confirmed that AlphaFold2, although trained only on single proteins with a folded structure, can be used as an intrinsic disorder predictor by repurposing low-confidence residue predictions[17–19]. AlphaFold2 low-confidence predictions on protein surfaces might also be indicative of possible binding regions[20,21].

Very soon after its release, AlphaFold2 was also tested for its capacity to predict protein-protein interactions. Despite not being designed for this purpose, AlphaFold2 outperformed traditional methods for the structural prediction of complexes between globular protein domains, in terms of both success rate and model quality[22–28]. AlphaFold-Multimer, specifically retrained on protein complexes, displayed improved performance for interface modeling over the original AlphaFold2[22,23,29]. At a wider scale, a systematic exploration of the yeast interactome used prefiltering with a fast version of RoseTTAFold[30] followed by AlphaFold2 structure prediction[31]. This opened exciting perspectives for the use of AlphaFold2 not only for complex structure prediction, but also as an in silico screening tool for interactions, as recently illustrated by the discovery of DONSON's role in replication initiation[32].

AlphaFold2 predictions are sensitive to the input parameters, input MSA and protein delimitations. For instance, the quality of structural models could be significantly improved using the AFsample strategy, relying on the massive generation of up to 6000 models using different sampling schemes[33]. AlphaFold2 can also be made to predict alternative conformational states for some proteins through manipulation of the MSA[34] either by subsampling[35] or by in silico mutagenesis[36]. For complexes, the generation of a paired MSA, where species are matched between homologs of the different protein partners, was not found to be necessary for AlphaFold2 to pick up interaction signal[22,26], although combining unpaired and paired MSAs gave the best results[22]. The AlphaPulldown package allows users to select or screen protein fragments for modeling protein complexes, since some interactions cannot be predicted if the full-length sequences are provided to AlphaFold2[37].

Interactions mediated by short peptides within disordered protein regions are quite specific and thus require extra care for handling by AlphaFold2. Indeed, conformational versatility is even higher and covariation signal is weaker than for globular complexes[38]. Traditional tools to predict protein-peptide interactions include mostly docking approaches, recently reviewed in[39,40]; some of these also make use of evolutionary information[13]. Several recent studies have addressed the ability of AlphaFold2 to predict protein-peptide complexes. An early implementation already showed interesting predictive capacity, including in cases where the peptide induces a large conformational change of the protein and docking therefore most likely fails, and without the need for a peptide MSA[41]. InterPepScore[42], a graph neural network used to score protein-peptide complexes for improving Rosetta FlexPepDock refinement[43], was also found beneficial to refine AlphaFold-Multimer models. AlphaFold-Multimer performs better than AlphaFold2 at protein-peptide complex prediction[44], and sampling a larger part of the conformational space by enforcing dropout at inference time in AlphaFold-Multimer further increased the quality of protein-peptide complex models[45]. Finally, the importance of choosing the right delimitations to optimize the sensitivity of the AlphaFold2 predictions has recently been highlighted for a number of protein-protein interactions involving disordered regions[46].

In the present study, we investigate how best to use AlphaFold2 to make the leap from interaction networks to interfaces when dealing with binding partners containing intrinsically disordered regions (Fig. 1a). We carefully develop an unbiased benchmark of 42 protein-peptide complexes sharing no similarity with any complex from the AlphaFold-Multimer training dataset and assess the performance of AlphaFold-Multimer on this dataset using different MSA schemes. We show that performance is limited when full-length protein sequences are used as input and considering delimited fragments increases the success rate. We set the fragment size at 100 amino acids in order to scan potential interacting regions within full-length sequences such as those derived from large-scale interactome data. The fragment scanning approach on the 42 protein ligands reveals that in 89% of the cases the fragment with highest ipTM score matches the region containing the correct binding site. Once the correct delimitations are identified, we show a synergistic effect of combining different MSA schemes and scores, reaching more than 90.5% success rate on our benchmark dataset. We also observed this synergy when using a larger dataset of 923 protein-peptide interactions extracted from the ELM database. Finally, our study also raises the issue of prediction specificity, which may require the enumeration and ranking of potential anchoring sites, and assesses the usefulness of AlphaFold confidence scores in discriminating between possible binding regions.

## Results

### Selecting a test dataset of complexes not redundant with the training set of AF2-Multimer

To assess the performance of AlphaFold2 (AF2) in predicting the mode of association between a protein (hereafter called the receptor) and a small binding motif within a structurally disordered partner (the ligand), it is important to study cases of complexes that do not have homologs in the database on which AlphaFold2 has been trained. An example of how AF2 models may be biased by existing structures in the PDB is illustrated in Supplementary Fig. 1a. AF2-Multimer was trained on structures released until 30 Apr 2018. An analysis of the structures released after that date revealed that nearly 2,500 structures of complexes involving small protein motifs had been deposited in the PDB (Fig. 1b). A large number of these structures have significant similarities in sequence or structure with structures released in the PDB before May 2018. Following a strict treatment of this sequence and structure redundancy (Fig. 1c, see Methods), we isolated a set of 42 complexes involving a receptor and a small peptide ligand that could provide an unbiased estimate of AF2 performance in different conditions (Supplementary Table 1). Among the 42 complexes, we observed a diversity of subunit lengths (Fig. 1d) and a representative occurrence of peptides with sizes ranging from 6 to 39 amino acids (Supplementary Fig. 1b) that are binding their receptors as helices, strands, coils or combination of those (Supplementary Fig. 1c).

AF2 relies on multiple sequence alignments whose evolutionary depth on the ligand peptide region may be limited due to the difficulty of identifying homologs from a short IDR sequence. Hence, for each of the proteins in this dataset, we used the full-length sequences of the protein partners to construct MSAs and subsequently delineate the interacting domains (Supplementary Fig. 2). These MSAs were combined to generate mixed co-alignments in which partner sequences belonging to the same species were paired while those with a single partner homolog present in a species were added as unpaired, similarly to ref. 47 (see Methods). When the receptor and ligand are considered in their integrality, the overall length of the concatenated sequences is in majority between 1000 and 2000 amino acids, significantly larger than when the size of the inputs is delimited to the boundaries used for structural determination (Fig. 1d). As a first analysis, we assessed whether AlphaFold2 was able to identify the correct binding site when proteins were considered in their full length. This is typical of a

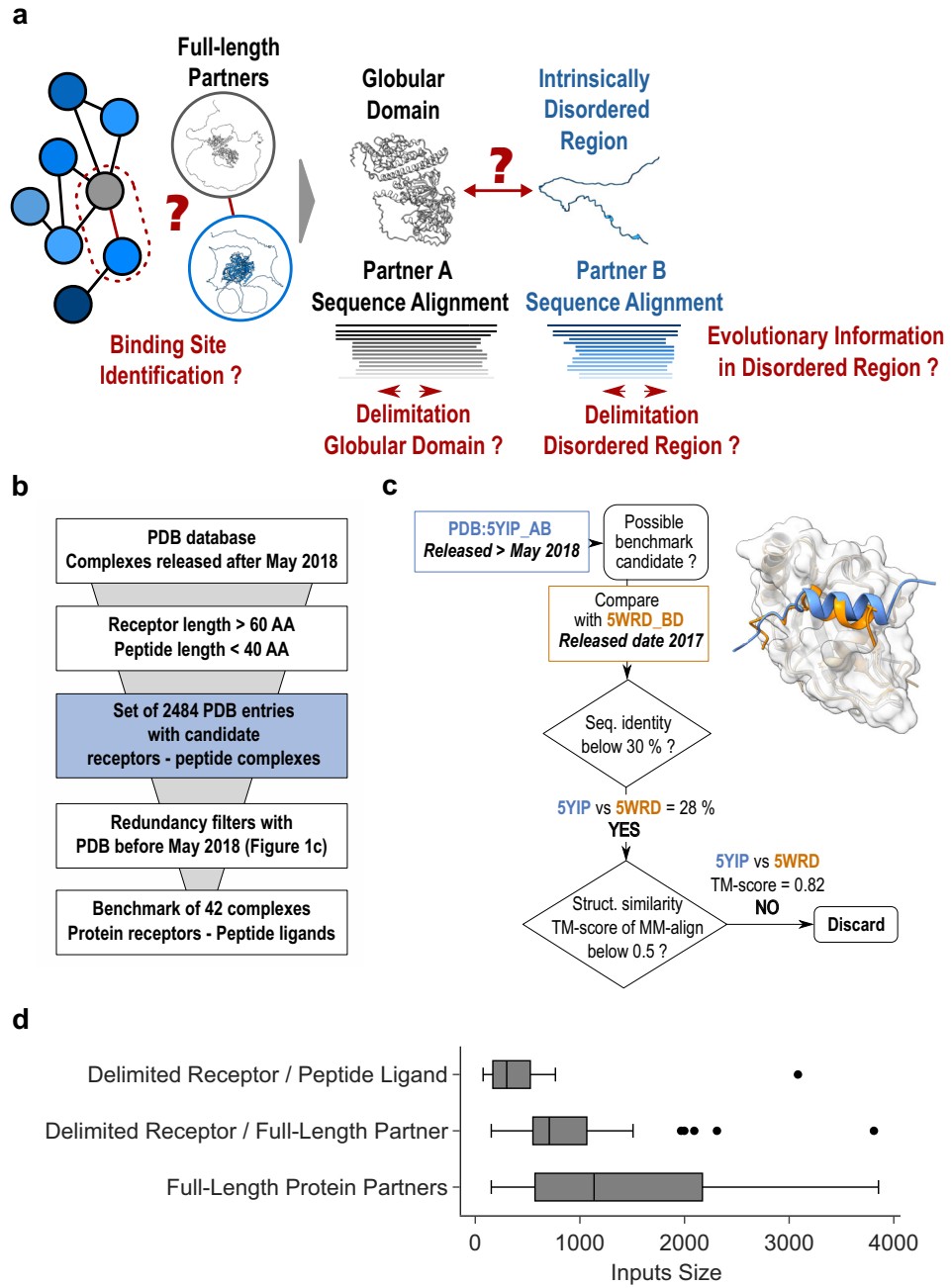

**Fig. 1 | General presentation of the benchmark dataset. a** Disentangling the complexity of a protein interaction network (sketched on the left) by analyzing binary interactions between a central gray protein and its blue binding partners can be complicated in case they contain intrinsically disordered regions. **b** General pipeline to select the PDB entries that can be used as test complexes from those released after May 2018. They were required to share no sequence or structural redundancy with any of the complex structures that were used for AlphaFold2-Multimer training. **c** Example illustrating filters used to assess the lack of redundancy between the candidate complex and structures published before May 2018. Two filters were used, one based on sequence identity using a 30% seq. id. threshold and a second retrieving all complexes involving a receptor homolog using PPI3D[57] and checking for lack of interface structural similarity using MM-align[59]. **d** Boxplots showing the cumulative size distribution of the 42 inputs (receptor+ligand) that were processed by AlphaFold2, either in protocols where sequences were delineated following the boundaries of the experimental structures or in those where full lengths of ligands and/or receptors were used. In the boxplot representation center line is the median, min and max limits of the box are the lower and upper quartiles, whiskers are the 1.5x interquartile range and points represent outliers. Source data are provided as a Source Data file.

scenario where knowing that two proteins are interacting, we have no initial indication of which regions are involved.

## Success rates of AF2-Multimer for full-length and delimited input protein partners

For each run, 25 models were generated with AF2-Multimer parameters, following the reference protocol[29]. The AF2 model confidence score (noted AF2 confidence score below), consisting of an 80:20 linear combination of ipTMscore and pTMscore, was used to rank the models and identify the best model. The accuracy of this best model was used to calculate the overall success rate for the 42 cases using the stringent criteria defined by the CAPRI community to assess the precision of protein-peptide complex models (see Methods and Supplementary Data 1 for detailed scores). With full-length protein partners,

we obtained a success rate of 42.9 % (Fig. 2a), rather low with respect to that reported in the evaluation of AF2-Multimer for protein complexes, which was benchmarked using delimited sequence inputs[29]. Analysis of the quality of the best model as a function of the size of the modeled assembly (Supplementary Figure 3a) shows that the performance tends to decrease for large sizes above 1600 amino acids although it is still possible to observe good predictions above this size threshold. Below 1500 amino acids, the success rates do not appear correlated with the size of the assembly or the nature of the peptide secondary structure (Supplementary Figure 3a).

Next, the sequences of each binding partner were delimited according to their boundaries as observed in the experimental structure of the complex (Supplementary Fig. 2). When both the receptor and ligand were delimited, the overall success rate was much higher, reaching 78.6% of the 42 complexes correctly predicted (Fig. 2a). In these first tests, the evolutionary information was integrated using the mixed co-alignment mode described above. We also tested alignment conditions in which the co-MSA is constructed from the same sequences but concatenated as an unpaired alignment (without matching homologous sequences between partners). In this unpaired mode (Supplementary Fig. 2), the success rate remained similar at 78.6% (Fig. 2a), suggesting that homolog matching in the paired alignment does not provide a major gain. We assessed a third prediction mode in which no evolutionary information is added in the peptide region, as performed in refs. 41,45 (Supplementary Fig. 2). With this third approach, the performance obtained without evolutionary information on the peptide side remains high, with 71.4% of correct models for the 42 cases (Fig. 2a).

Such a good performance in the absence of any alignment associated with the peptide confirms that the properties of the binding site in the receptor are often sufficient to guide the interaction mode of the peptide[41,45]. Consistently, in a situation where the receptor is delimited but the ligand is considered in its full-length sequence, the performance drops back to a lower level of 52.4%, even when using the MSA information on the ligand side (Fig. 2a, Supplementary Fig. 3b). Hence, one of the difficulties encountered by AF2 in dealing with large IDR-containing proteins lies in its ability to identify the correct interaction region within the partner protein.

The success rates calculated above were obtained by selecting only the model with highest AF2 confidence score among the 25 sampled models. Considering the entire set of 75 models (25 models for every complex in the three alignment conditions: mixed, unpaired, no_ali) highlights a significant Pearson's correlation of 0.84 between the AF2 confidence score and the DockQ score, a commonly used metrics to rate the accuracy of modeled interfaces with respect to the reference complex[48] (Fig. 3a). Grouping the models according to their CAPRI quality ranks (Acceptable/Medium/High) (Fig. 3b) using the stringent criteria used for protein-peptide complexes[49] (see Methods) highlights that above an AF2 confidence score of 0.65, the predicted models are most often correct. There is also a minority of cases with a score between 0.4 and 0.65 that are found correct (in the Acceptable category) indicating that this twilight zone may be interesting to investigate if no alternative solution has been detected. In any case, the graphs on Fig. 3a, b confirm that the AF2 confidence score (see Methods) can be used as a reliable proxy for estimating the reliability of a protein-peptide interaction prediction.

## Success rates of AF2-Multimer considering protein fragments of increasing size

When searching for an interaction site between two proteins, the region involved in the interaction is usually not known precisely. In order to use AF2 to carry out this task, and given the lower performance of AF2 with full-length proteins, we explored how AF2 predictions would be impacted by queries in which the bound motif is not perfectly delineated and is embedded in a larger fragment that may

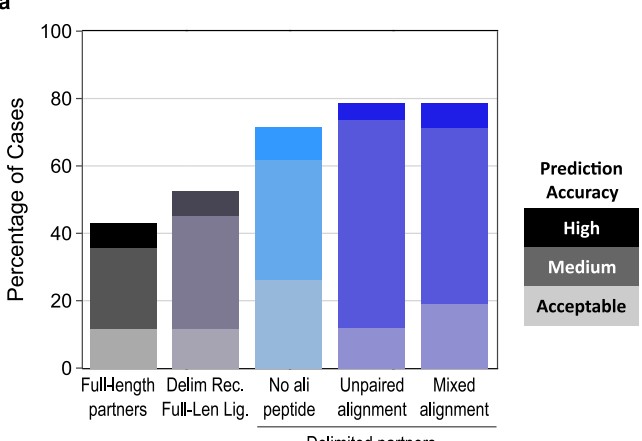

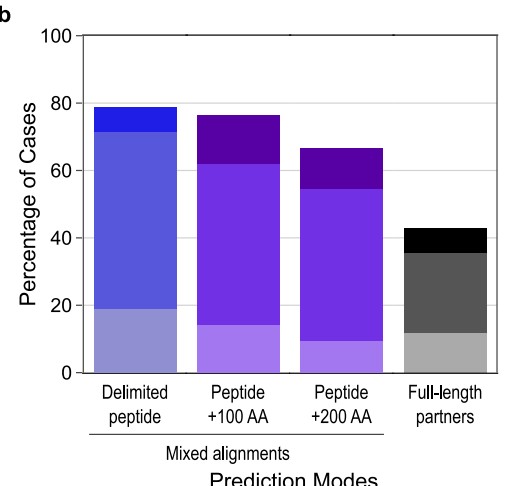

**Fig. 2 | AlphaFold2-Multimer success rates on the benchmark dataset using different prediction modes.** Stacked barplots reporting the success rates of AlphaFold2 prediction depending on the types of co-alignment used. All success rates are presented as the percentage of test cases in which the best AF2-Multimer model (best AF2 confidence score) is of Acceptable (light color), Medium (medium color) or High (dark color) quality according to the CAPRI criteria for protein-peptide complexes[49]. **a** Success rates using (from left to right): full-length partners with a mixed alignment generation mode (gray grades), delimited receptor with full-length ligand with a mixed alignment generation mode (deep purple grades), delimited partners with no evolutionary information for the peptide (cyan grades), delimited partners with unpaired co-alignment (blue grades), delimited partners with mixed alignment (blue grades). **b** Success rates using (from left to right): delimited partners (blue grades) (same as rightmost bar in panel **a**), peptides extended by 100 or 200 amino acids (purple grades), full-length partners (gray grades) (same as leftmost bar in panel **a**). Source data are provided as a Source Data file.

contain 100 or 200 additional amino acids. Extending the sequence containing the binding motif of each complex with up to 100 or 200 amino acids, and delimiting the alignments constructed in a mixed alignment mode (Supplementary Fig. 2), we obtained a decrease by 2.4 to 11.9 points with success rates of 76.2% and 66.7%, respectively for fragment size 100 and 200 (Fig. 2b). The success rate of 66.7%, obtained for cases where the fragment extends the peptide motif by 200 amino acids, is substantially higher than the 42.9% obtained with full-length proteins. This result underscores the interest of fragment-based searching to identify potential interaction motifs between two partners and to predict their recognition mode. Previously (Fig. 2a), we showed that the lack of evolution for the peptide was not very detrimental for a significant number of correct predictions (71.4%). This

**a**

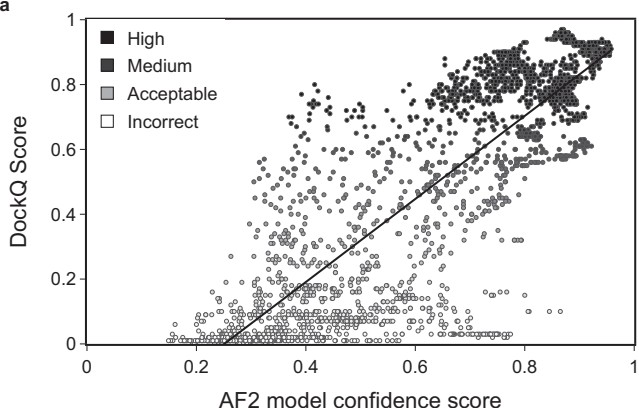

**b**

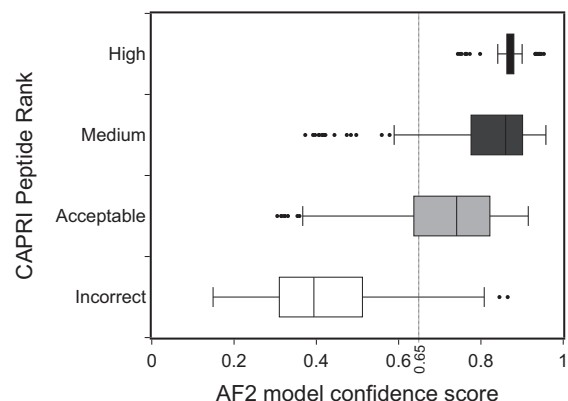

**Fig. 3 | Model quality depending on the value of the AF2 model confidence score. a** Distribution of DockQ scores[48] for 75 models for every binary protein-peptide complex (25 models in the three alignment conditions: mixed, unpaired, no_ali) as a function of the AF2 model confidence score. Data points are colored according to the model quality as rated by the DockQ score from low (white) to high (dark gray) values. Pearson's correlation is 0.84. **b** Boxplots of the AF2 confidence score value distributions for the same set of models, split by model quality category according to the CAPRI protein-peptide criteria: High (sample size $n = 128$, dark gray), Medium ($n = 332$, medium gray), Acceptable ($n = 106$, light gray), Incorrect ($n = 484$, white). In the boxplot representation center line is the median, min and max limits of the box are the lower and upper quartiles, whiskers are the 1.5x interquartile range and points represent outliers. Source data are provided as a Source Data file.

trend is less pronounced when using fragments extended by 100 or 200 amino acids as shown in Supplementary Fig. 4a. Without ligand alignment, there is a loss of performance of more than 20 points, which highlights the importance of associating evolutionary information when the binding site identification involves a systematic search within larger fragments. For fragments of length 200, without evolutionary information for the ligand, the success rate is 45.2%, almost as low as the success rates obtained for full-length proteins with evolution.

### Success rates of AF2-Multimer when scanning a binding partner with overlapping fragments of fixed size

The success rates obtained with extended fragments (Fig. 2b) prompted us to assess whether AF2-Multimer would be suitable for screening and ranking different overlapping fragments along the sequence of a binding partner. To do this, we considered each of the 42 pairs of binding partners in the benchmark dataset, delineated the receptor binding domain and generated models of the complex between this receptor domain and every fragment of the ligand protein of size 100 amino acids with overlaps of 30 amino acids between

fragments (Fig. 4a). The models were ranked according to their ipTM scores (Fig. 4b) and discriminated based on the overlap of the fragment with the binding site motif.

For 35 cases in which the ligand protein was larger than 100 amino acids, the fragment with the highest ipTM score overlapped with the correct region of binding in 31 cases (89 % success rate) (Fig. 4c and Supplementary Data 2). Four cases were incorrectly predicted, i.e. one fragment not overlapping the binding site had the highest ipTM score. In three of these four cases, the correct models had low ipTM scores. However, the fourth case, 39_7O6N, was incorrectly predicted despite the correct model having an ipTM score of 0.646. In PDB 7O6N, the receptor is in the form of a dimer. Since its ligand binds as a helix away from the dimer interface, the receptor was modeled as a monomer in the benchmark (Supplementary Fig. 5a). However, the surface of the receptor involved in homodimer formation tends to create an interaction surface that traps the 100 amino acid long fragments and generates ipTM scores higher than that of the correct interface (Supplementary Fig. 5d, e). When the ligand was delimited as in the PDB, AF2-multimer managed to predict the correct interface even with the receptor as a monomer (Supplementary Fig. 5b, c). This example highlights the importance of modeling the receptors with as many permanent binding partners as possible (either as homomers or heteromers) to prevent large hydrophobic surfaces from misleading AF2-multimer predictions with extended fragments containing short linear binding motifs.

### Advantage of combining different alignment modes

The performance obtained using different alignment modes and input lengths suggests that some complexes can be correctly predicted regardless of the protocol used, while others may be sensitive to these input conditions. Overall, for 35.7% (15 complexes), a correct model could be ranked first using the AF2 confidence score with any of the input conditions, even using full-length alignments (Supplementary Fig. 6). In contrast, other complexes could only be predicted correctly with a limited set of conditions (Fig. 5a), suggesting a potential interest for combining different strategies. Instead of considering 25 models generated with every protocol, we analyzed a pool of 100 models generated with four different protocols and ranked them according to the highest AF2 confidence score. The resulting success rate improves significantly, rising up to 90.5% (Fig. 5b). The AF2 model confidence score is sufficiently correlated with the accuracy of the models that it can be used to identify correct assemblies in much larger model sets. We verified that sampling 100 models rather than 25 did not change the success rates of single protocols, whereas generating 100 models through a combination of four protocols increased the success rate by almost 12 points, highlighting the value of increasing the sampling by varying the properties of multiple sequence alignments (Supplementary Fig. 4b).

In an attempt to interpret the failures and successes of the tested protocols, we performed a detailed analysis of complexes that specifically succeeded with only a subset of the protocols and those that did not succeed with any. A typical case is when the conformation of the bound peptide is best predicted in the absence of evolutionary information. The absence of evolutionary information was found to be favorable for complexes such as 6ICV or 6YN0 that were not correctly predicted when MSA was added to the peptide. In the case of 6ICV (Fig. 6a), the peptide (blue) is predicted to adopt a helix-and-turn conformation with high confidence when the evolutionary information of the MSA is included. This local structure is incompatible with the extended bound conformation. In contrast, in the absence of evolutionary information, the predicted structure of the peptide (light blue) is in very good agreement with the experimental structure (red), suggesting that the geometric constraints have been relaxed sufficiently for the peptide to sample an extended geometry that was well evaluated by the AF2 confidence score.

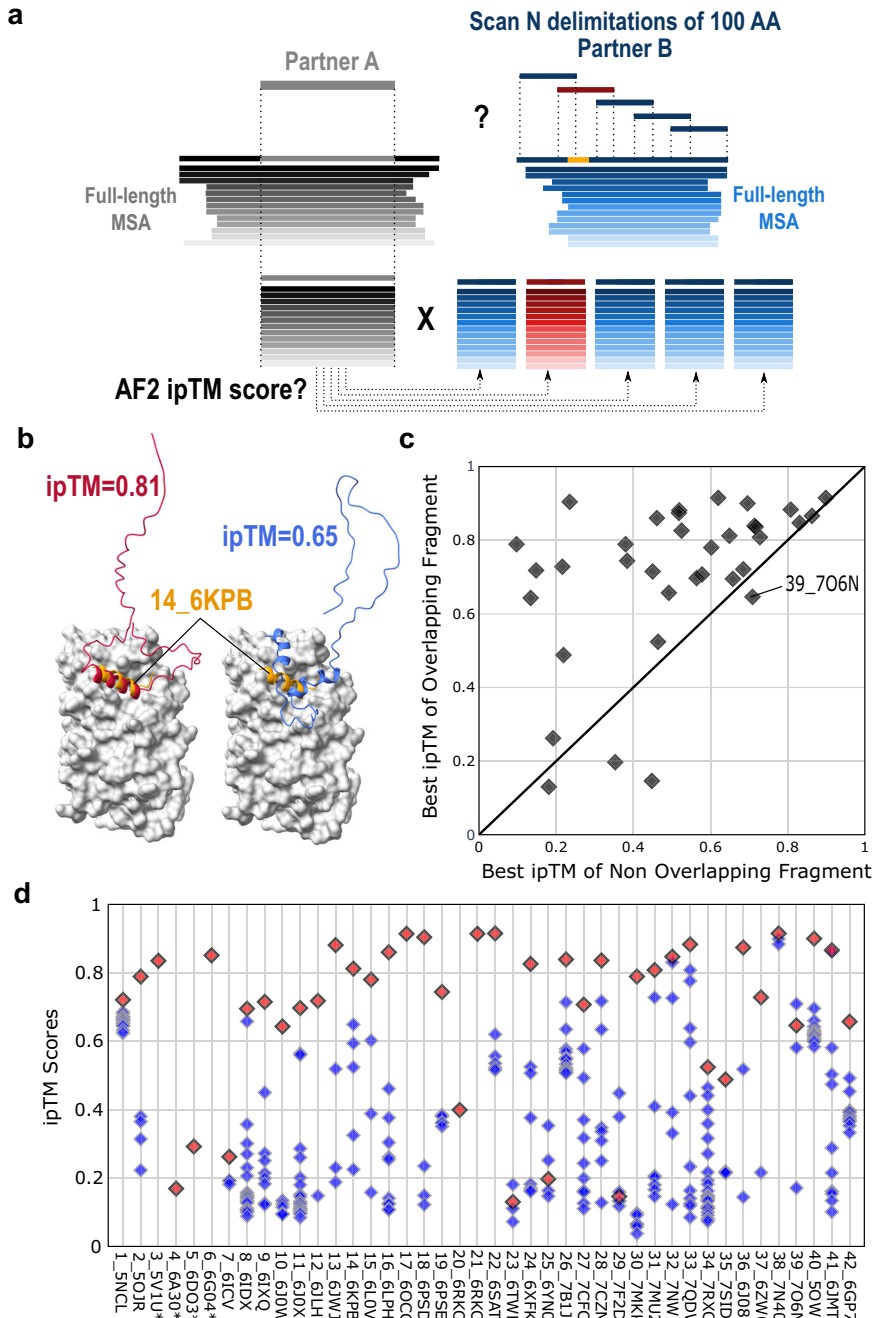

**Fig. 4 | Systematic screening of the different fragments of a binding partner.** **a** Protocol used to screen the complete ligand sequence of a binding partner by analyzing all 100 amino acid long fragments against the receptor delimited by the length of its interaction domain. The fragment overlapping the correct binding site shown in orange is colored red while the other fragments are blue. **b** The predicted ipTM score is used to rank the different fragments and evaluate those that overlap or not with the correct binding site. **c** Scatter plot showing the highest ipTM score for the model containing a fragment overlapping the correct binding site compared to the highest ipTM score among models with no overlap. 35 points are represented and not 42 since 7 ligands have less than 100 amino acids. **d** Detailed distribution of ipTM scores for the 42 PDB cases of the benchmark with the fragment overlapping the correct binding site as a red diamond and the non-overlapping ones as blue diamonds. If two fragments overlap with the binding site, only the model with highest ipTM score is represented in panels **c** and **d**. Source data are provided as a Source Data file.

Other differences between the tested protocols could be observed in case a motif, well predicted in a short fragment, was not correctly predicted in longer ones. This is observed in 5 cases including PDB cases 7F2D, 6ZW0 and 6JLH illustrated in Fig. 6b–d, respectively. For these systems, considering the ligand peptide in the context of a larger fragment with 200 additional amino-acids (dark blue models) never led to a correct prediction by AF2, while the delimited peptides (blue) were always modeled in good agreement with the experimental reference structure (red). In almost all of these complexes, the origin of the failure in the larger fragments seems to be due to the presence of intramolecular contacts involving the peptide and surrounding regions. In the case of 7F2D and 6ZW0, the peptide is located in the vicinity of a globular domain with which it forms contacts of relatively low confidence. However, these appear to be sufficient to interfere with the generation of the native complex. In the third case, 6JLH, the binding peptide is embedded in a longer coiled-coil that masks the surface found to bind the receptor experimentally. This prediction would be consistent with the experimental study that showed the

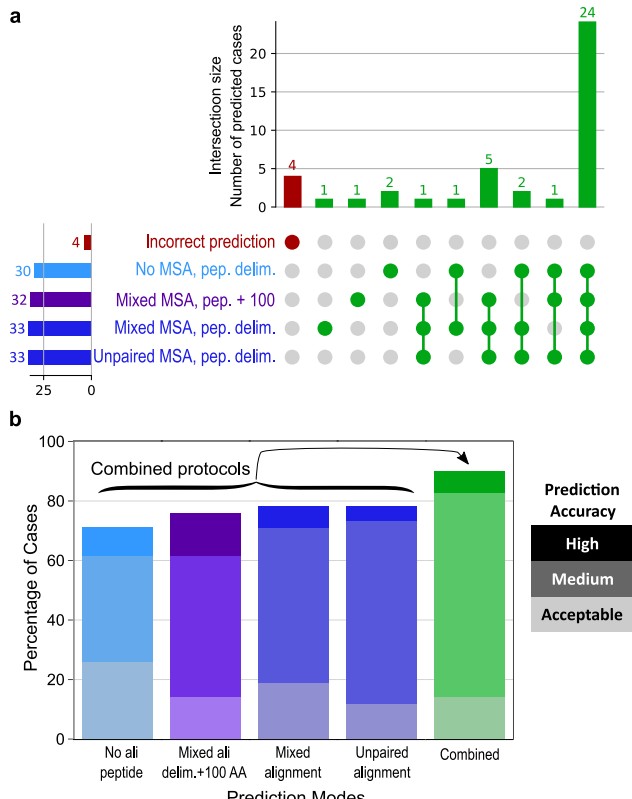

**Fig. 5 | Complementarity of the predictions made in different prediction modes. a** UpSet diagram[69] displaying the number of successful cases (out of 42) found by either a single or several prediction mode(s) among the following: delimited peptide with no peptide multiple sequence alignment (MSA), peptide extended by 100 residues with a mixed MSA, delimited peptide with a mixed MSA, delimited peptide with an unpaired MSA. 5 cases that can be identified in none of these conditions are highlighted in red. **b** Success rates for the four protocols shown in panel a (values are the same as in Fig. 2) and for a combined protocol taking the best AlphaFold2 (AF2) confidence score value out of 100 models (25 for each condition) (green grades). Source data are provided as a Source Data file.

interaction to be observed only in specific physiological contexts[50]. This example together with another case also involving long coiled-coils (7MU2) highlights the value of exploring different fragment lengths to reveal the appropriate binding epitopes. Therefore, in the cases where prediction performance varies between alignment content and delineation protocols, a common explanation is that the binding motif may be trapped or masked in a conformational state that prevents prediction of its correct binding mode.

Four cases out of 42 failed, regardless of the alignment protocol. In one of these cases (PDB: 7CZM), the receptor itself was not quite well folded, which may have made it difficult to sample a correct binding mode. For one case (PDB: 6A30) where none of the tested protocols converged to a correct model, we tested whether reducing the size of the receptor itself would help. We reran this case with the same alignments, testing if reduction in the size of the receptor could have an impact. Splitting the receptor as two inputs of similar size led to the generation of a correct model with a high AF2 confidence score with one of those inputs, reaching 0.8 when using the protocol with no peptide alignment but below 0.5 with all the other protocols. With this additional complex, the percentage of cases that could be predicted using AF2 rises above 92%. Hence, there is room for further improvement by sampling simple alterations of the input MSAs and using the AF2 model confidence score as a guide for identification of the correct protocol.

## Specificity for similar binding motifs recognized by receptors

Out of the 42 cases in the test set, AF2 is able to correctly predict the binding mode of a peptide to its receptor without any evolutionary information for the peptide in 71.4% of the cases. Such a performance suggests that the structural and evolutionary properties of the receptor match well with the peptide sequence irrespective of its conservation pattern. This calls into question the ability of AF2 to distinguish cognate binding peptides from non-binding ones. This issue may be particularly difficult in the challenging cases where two short fragments embedded in long disordered regions of different binding partners need to be discriminated while they tend to adopt a similar local conformation. To address this issue, we distinguished different classes of complexes based on the secondary structure adopted by the peptide in its bound conformation in order to create 83 challenging cross-partners predictions between 23 receptors and cognate or non-cognate ligands selected from the 42 cases of our test set (Supplementary Table 2). We then assessed whether AF2 could specifically predict the binding mode of receptors with their respective peptides and distinguish them from potentially misleading peptides taken from unrelated structures but sharing similar bound conformations.

In total, 7 categories of peptide conformations were considered (Supplementary Fig. 7). We distinguished those binding through a small, medium, or long helix, those showing no canonical secondary structure and those binding through the formation of a combination of helix and strand or a single or two beta-strands (Supplementary Fig. 7a–g). To run the cross-partners interaction tests, we used the protocol with no MSA in the peptide region. Over the 23 selected cases for cross-partners analysis, 16 were successfully predicted by AF2 (70%) in agreement with the performance obtained with this protocol on the 42 test cases. In Fig. 7a, the distribution of AF2 confidence scores obtained for the models rated as correct using the CAPRI protein-peptide criteria (darker blue distribution) differs significantly from the distribution of the scores obtained with non-native peptide ligands (light blue distribution). The AF2 confidence score of the specific peptide was superior to any of the non-specific peptides in 15 out of the 16 correctly predicted complexes. Figure 7b illustrates one of these 15 cases, using the receptor of 7CFC, highlighting that even if the non-specific peptides tend to interact in the same region as the specific one, the AF2 confidence score is higher for the specific peptide (reaching 0.75) and can be used as a proxy to discriminate between several likely binders.

Based on the distribution in Fig. 7a, a minority of models (approximately 10%) would give a misleading assignment for AF2 confidence scores greater than 0.6 and could prevent identification of a correct binding site. This is illustrated by the case of the 6IDX complex in Fig. 7c in which an incorrect binder (6KPB Ligand) is predicted to form a complex with the receptor with high confidence (as indicated by an AF2 confidence score of 0.83) while the specific ligand was not correctly predicted (AF2 confidence score = 0.46 and wrong binding mode).

Last, there are also alternative situations as illustrated in Fig. 7d in which the score of the specific binder is mild (below 0.6) but still among the highest scores obtained in the set of potential binders. This was observed for 3 of the 16 cases where the specific receptor-ligand pair was correctly predicted by AF2 (6YN0 in Supplementary Fig. 7c and 7F2D, 6J0W in Supplementary Fig. 7f). With the 6J0W receptor (Fig. 7d), the AF2 confidence score of the specific ligand is 0.53, whereas it reaches 0.54 with another non-specific ligand of 7CZM. Such a situation highlights the specificity issue that may arise in the case where the peptide is not accurately modeled in the receptor binding site. It can be noted in Supplementary Fig. 7f that the misleading 7CZM ligand tends to have higher AF2 confidence scores than the other ligands on average with all the non-specific receptors. Such promiscuity indicates the risk that some sequences may systematically

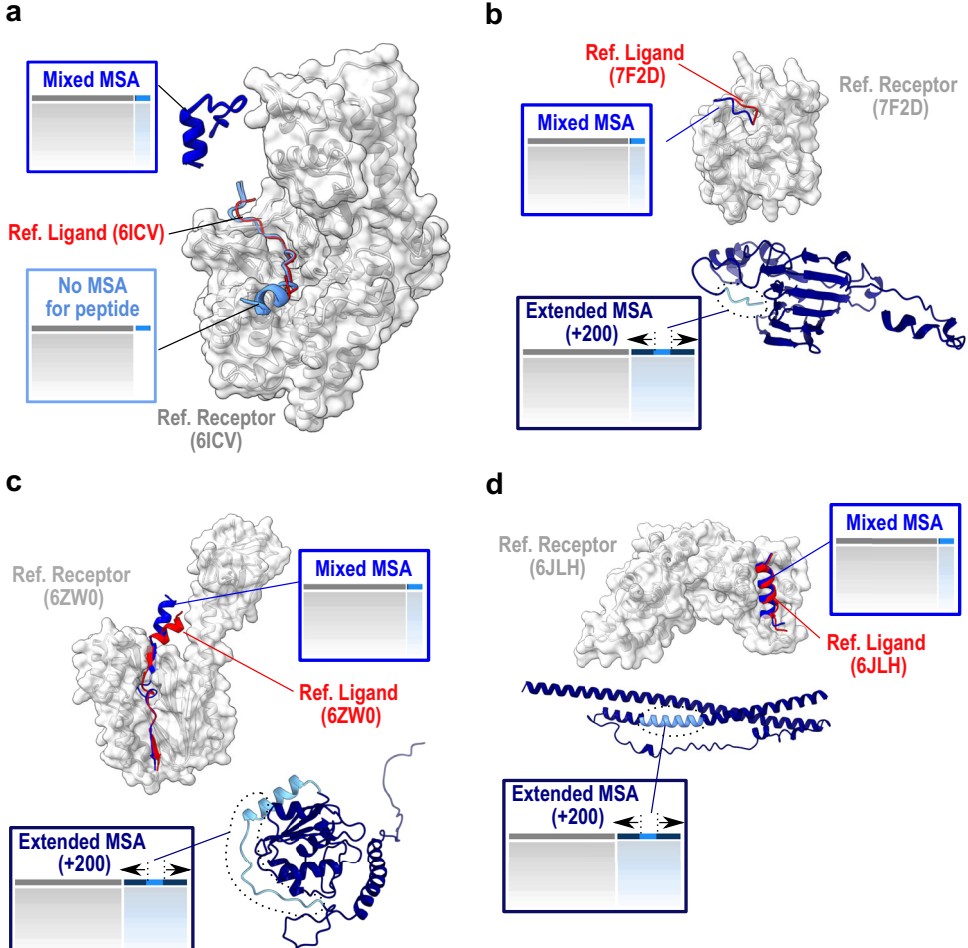

**Fig. 6 | Detailed analysis of complexes that succeed with only a subset of the protocols.** The receptor is represented as a gray surface, the native ligand as a red cartoon, the predicted peptides in shades of blue: bright blue for the predictions in mixed multiple sequence alignment (MSA) mode, and light blue for the prediction with no peptide MSA (**a**) or for the peptide within the prediction of a fragment extended by 200 residues in dark blue (**b**–**d**). PDB identifiers of represented cases are 6ICV (**a**), 7F2D (**b**), 6ZW0 (**c**) and 6JLH (**d**).

bias the specificity analysis and that normalization or the use of an alternative scoring scheme might be useful to further disentangle the complexity of protein-protein interaction networks involving unstructured regions. For a few specific classes of binding motifs, a recent comparison with experimental data also indicated a lack of specificity for AF2 predictions[51]. However, in the absence of further biophysical experiments, we cannot completely rule out that, for the misleading assignments discussed above, non-specific peptides may indeed exhibit detectable binding to their non-cognate receptors.

**Extension to a larger dataset from the Eukaryotic Linear Motif database**

In the previous sections, we assessed the value of the fragment scanning strategy using a dataset of 42 receptor/ligand pairs, ensuring that this assessment was unbiased with respect to AF2 training process. One drawback is that this dataset is limited in size, and we wondered whether the performance of the method would be maintained if we used data from the larger Eukaryotic Linear Motif (ELM) database[4] with the risk that the predicted cases would be biased by their similarity to the cases used in the AF2-Multimer parameter training. The ELM database contains a very large number of binding motifs identified in the disordered regions of proteins. These ligands are generally identified based on a consensus sequence motif established by experimental characterization of the interaction specificities of the protein domains specialized in recognition of these linear motifs. For many of the linear motifs listed in the ELM database, an experimental reference is provided to validate the existence of the binding motif. We extracted a list of 1884 receptor/ligand pairs with defined delimitations from the linear motifs associated with a reference in PubMed on July 3, 2023 (Supplementary Data 3). Among these pairs, the subset possessing (i) a unique binding site in the ligand and (ii) a PDB reference (either exact or homologous) contains 923 cases divided into 84 categories of ELM types, corresponding to different families of domains and their associated consensus motifs (Fig. 8a) (see Methods). The different protocols discussed above for using AF2 were applied to assess AF2's ability to model the bound motif correctly. To correct for the unbalanced distribution of ELM motifs within the 84 categories, we evaluated the predicted success rates by repeated stratified sampling with 1000 repeats of randomly selecting one ELM motif from each of the 84 categories.

We evaluated six protocols using an exact or a homologous PDB structure to rate model accuracy (see Methods) and the results are presented as a histogram in Fig. 8b and detailed in (Supplementary Data 3) (detailed performance with respect to each ELM types are provided in Supplementary Data 4).

The first protocol evaluates predictive performance in the case where the ligand is considered to be full-length and the receptor is delimited around the binding domain (see Methods). For all 84 categories, the repeated stratified sampling procedure on the 923 ELM motifs converges to a success rate of 52.7 ± 4%, very similar to the value

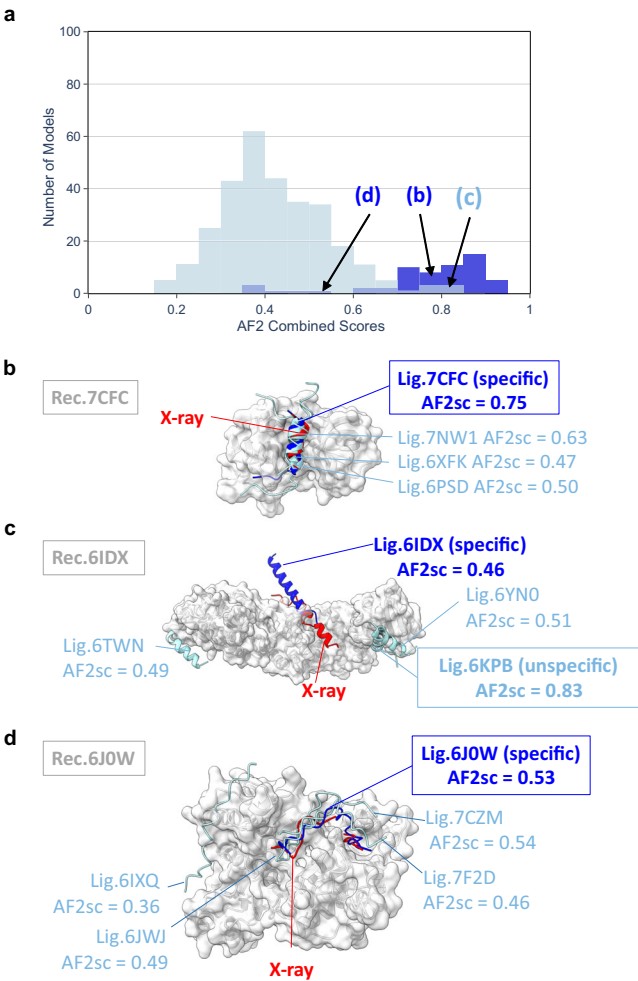

**Fig. 7 | Cross-partners evaluation assessing the specificity of the binding predictions. a** Distribution of AF2 confidence scores obtained for the models involving the native peptide and rated as correct using the CAPRI protein-peptide criteria (darker blue distribution) for 16 out of 23 cases selected for cross-partners evaluation and for the models obtained with non-native peptide ligands (light blue distribution). Cross-partners predictions were performed using delimited partners with no evolutionary information for the peptide. Specific predictions illustrated in panels **b**–**d** are drawn from the relevant distributions at the indicated score values. Illustration of specific cases discussed in the text for native PDB identifiers: 7CFC (**b**), 6IDX (**c**), 6J0W (**d**). The receptor is shown as a gray surface, the native peptide as a red cartoon, the best predicted model involving the native peptide in bright blue cartoon and the best predicted models involving non-native peptides in light blue. AF2sc values indicate the best AF2 confidence score values obtained for each complex.

binding motif is very short (3 amino acids). The specific success rates obtained for ELM cases for which the reference structure is an exact PDB structure (differentiated in Supplementary Fig. 8) are higher than those for which only a homologous PDB structure is known.

## Discussion

AF2 has shown remarkable performance for predicting the structure of multi-subunit machineries only known so far through PPI maps[27,28,31]. In this study, we explored the potential of AlphaFold2 to further exploit the wealth of data contained in proteomics experiments and to enable a more comprehensive characterization of protein-protein interaction networks. We focused on interactions mediated by unstructured regions that are a cornerstone of the functional and dynamic organization of most cellular processes. The capacity of AF2 to perform well with small disordered regions binding a structured domain was established on different datasets[18,41,45] built from the structures available in the PDB[52]. However, in most proteomics experiments, the boundaries of the interacting regions are not precisely known. In addition, physical interactions can be indirect and it is crucial to disentangle the regions involved in direct interactions.

To further assess the use of AF2 for this purpose, we built a dataset of complexes consisting of a folded receptor bound to a short protein fragment and evaluated several protocols representative of challenges faced following proteomics analyses. Because AlphaFold2-multimer was trained on complexes whose structures were published before May 2018, we carefully removed any homologs of the complex to ensure that our conclusions could not be biased by similarities in sequence or 3D geometry with the training dataset. For the 42 test cases selected in the benchmark, we first evaluated the ability of AF2 to discriminate the binding site when proteins are provided in their full length as in the output of a proteomic experiment. We achieved a rather low success rate of 42.9%, in agreement with the observations made in a recent report dealing with another set of interactions involving short linear motifs[46]. We noted that above 1600 amino acids, the method gave poorer predictions, with the exception of two impressive cases above 2500 amino acids. The use of input fragments delimited as in the experimental structures significantly increased performance by more than 35 points and the combination of different MSA construction modes led to an overall success rate of 90.5%. If the binding region is unknown, scanning multiple small peptides can be computationally demanding and we found that a reasonable trade-off in accuracy could be achieved with a fragment length of about 100 amino acids. We show that using a fragment scanning strategy with fragments of 100 amino acids overlapping by 30 amino acids, the correct fragment could be identified in 89% of the tested cases. This result indicates that in most cases, it is unlikely that a wrong competitive binding site may be found in a ligand protein with ipTM scores as high as the cognate binding fragment.

The length of the scanning fragment might need to be reduced if a very short binding motif of about 3 amino acids is expected, as we found in the ELM database analysis that some of these short motifs, especially when formed by polar residues, were more difficult to predict in longer fragments (Supplementary Data 5). To increase success rate of the fragment scanning approach, our study also underlines the importance of representing the biological context as closely as possible, taking into account the homomeric and heteromeric assemblies pre-existing the formation of an interaction with a disordered region. A recent in-depth study of the cohesin interaction network using AF2 also led to conclusions along the same lines[53]. When using fragments of size larger than 100, evolutionary information was key to reaching the best results and for larger fragments involving more than 200 amino acids, a decrease in performance was observed which could originate from intramolecular contacts that tend to mask the binding region or hinder the sampling of the bound conformation. Finally, scanning strategies taking into account the location of globular domains to

of 52.4% obtained for the dataset of 42 non-redundant cases (Fig. 2a). Similar to what was observed for the dataset of 42 non-redundant cases, success rates increase significantly when the size of the fragments used for prediction is restricted around the interaction region. Similar values to those obtained in Fig. 5b are observed, with the best performance being obtained when unpaired or mixed alignments are used (80.7 ± 3.0% and 77.9 ± 3.4%, respectively). Taking the best AF2 confidence score obtained among the four protocols, the success rate increases up to 87.3 ± 2.7% reaching a value close to the 90.5% obtained for the non-redundant dataset. The lower performance of the predictions for the protocol with a fragment size extended to 100 amino acids is discussed on an analysis of 30 representative cases in Supplementary Data 5: 20% of failures are due to the existence of several nearby consensus binding sites, and 50% belong to specific domain categories such as SH2, integrins or NRP domains, for which the

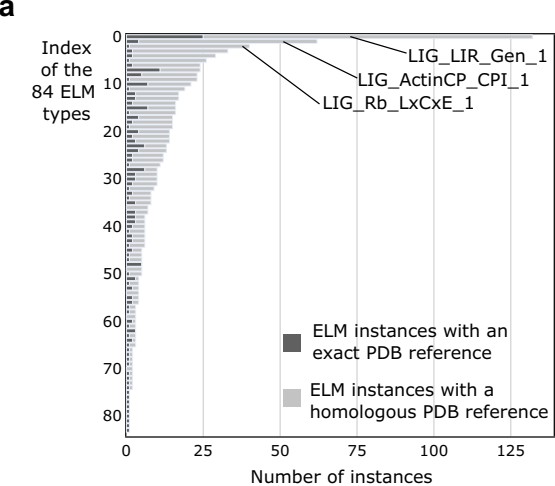

**a**

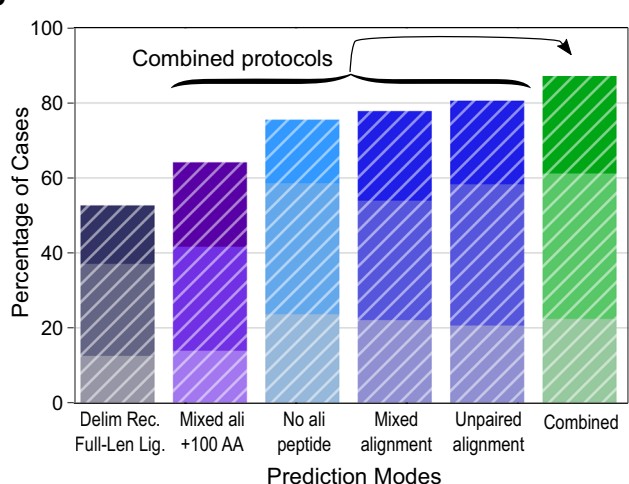

**b**

Fig. 8 | **Analysis of a large dataset of 923 complexes from the Eukaryotic Linear Motif (ELM) database. a** Distribution of the 923 ELM cases that could be modeled and evaluated among 84 ELM categories (ELM types), highlighting the unbalanced distribution, biased towards some receptor/ligand couples such as the LIG_LIR_-Gen_1, LIG_Actin_CPI_1 and LIG_Rb_LxCxE_1 categories comprising 132, 62 and 40 cases, respectively. Dark gray bars indicate the cases with an exact reference PDB structure while light gray bars indicate the cases that could be evaluated using a homologous PDB structure (see Methods). **b** Histogram representing the success rates obtained with six different protocols applied to the dataset extracted from the ELM database. The average success rates are reported with same color codes as in Figs. 2 and 5 but using white hatched bars (full-length ligand delimitation (deep purple), with extension of 100 amino acids (purple), with no alignment (cyan), mixed or unpaired co-alignments (blue), and combined (green)). The average success rates were calculated from a repeated stratified sampling of 1000 iterations over the 84 ELM categories, randomly selecting one ELM complex in each of the 84 categories. At every iteration, success rates were calculated and the mean success rate value is reported in the histogram where the best AlphaFold2 (AF2)-Multimer model (best AF2 confidence score) is of Acceptable (light color), Medium (medium color) or High (dark color) quality according to the CAPRI criteria for protein-peptide complexes[49]. Source data are provided as a Source Data file.

correctly delineate and cut the intrinsically disordered regions have also proven useful in increasing the success rate of predictions[46].

In the case of delimited peptides, it is remarkable that evolutionary information in the peptide region did not prove to be as crucial as for longer fragments for generating accurate models and scoring them reliably. We found that in specific cases where the bound conformation of the peptide was rather extended, the addition of evolutionary information was even detrimental to the identification of a correct solution. Such a detrimental effect of MSA was also reported in ref. 51 for the structural prediction of complexes between MHC receptors and various sets of short peptides by AF2. In these systems, the local conformation of the bound peptides is also fully extended. Our analysis suggests that the inclusion of the MSA for the disordered short peptide may lock the local conformations of the peptides and prevent them from adopting a different bound conformation. In any case, sampling these different possibilities was considered worthwhile, as the AF2 confidence score is sufficiently reliable to pick out the correct solution among those sampled. To further enhance the chance of generating a correct solution using short delimited peptides as ligands, we also explored how the AFsample strategy would perform on some of the difficult cases found among the 42 cases in our non-redundant dataset (see Methods). Our results highlight a complementarity of the two approaches, where some cases were successful with our combined protocol but not with AFsample, while others were unsuccessful in our combined protocol and solved by AFsample, albeit at a much larger computational cost (Supplementary Data 6a). A comparative analysis of the two approaches suggests a few guidelines that could be used to further increase success rates: the use of templates, of a combination of multimer_v1 and multimer_v2 parameters and a larger sampling for a given condition (up to 200 models per condition instead of 25). In contrast, on our dataset, using a larger number of recycles as in AFsample was never necessary to obtain successful predictions (using 9 or 21 recycles did not improve success rates). Additionally, if we had stopped sampling after 200 models for each condition instead of 1000 as implemented by default in AFsample, we would have obtained the same best models.

Beyond the remarkable ability of AF2 to generate correct conformations of protein-peptide complexes, we confirmed the reliability of the combined ipTMscore and pTMscore as an estimate of model accuracy. We also evaluated AF2 as a tool to discriminate a native ligand from other ligands potentially difficult to discriminate because adopting the same local conformation among diverse binding partners. The obtained results were satisfactory in a majority of cases where the AF2 confidence score correctly singled out the native binding peptide, but also highlighted several misleading situations that call for vigilance in the exploitation of specificity results. It certainly should be possible to reinforce the applicability of AF2 for the exploitation of more complex interactomes in which the interaction with unstructured regions plays a major role. Recent efforts in that direction have shown that AF2 parameters which were trained only with positive examples could be further fine-tuned for specificity combining positive and negative examples of receptor-peptide interactions[51]. So far, this fine-tuning was achieved in a receptor-specific manner focusing on MHC, PDZ or SH3 domains, but it might be expanded further to address other specificity issues.

The ability of AF2 to discriminate the native peptide from similar alternative binders when the native bound conformation is correctly predicted supports the conclusions that an energetic function of the protein structure may have been learned by AF2 independently of evolutionary information[54]. This ability to discriminate specific native binders is also consistent with the principle of using AF2 to discriminate peptide binders from competitive simulations[55] or for the design of high-affinity binders for their targets[56]. Using the strategy described in[55] could be a way to circumvent some specificity issues. Alternatively, rescoring complex models for different peptides using the updated AF2Rank program may provide complementary discriminative power[54]. Some receptors may also show more promiscuous binding properties than others when assessed from AF2 confidence score as shown in the case of 7CZM. Using a set of representative peptides such as those used in the present study, it may be possible to spot out receptors more prone to interacting non-specifically with various motifs and improve normalization of the confidence score. On the other hand, the fact that with larger

fragments (>200), the ability to identify the correct binding site decreases significantly and requires evolutionary information is also in agreement with the proposal that AF2 needs coevolution data to search for global minima in the learned function[54]. To progress from interactomes to the identification of all potential binding sites within disordered regions, a robust strategy will benefit from systematically scanning fragments of sequences of limited length and sampling different types of evolutionary information, such as the four combined in this study.

## Methods

### Building the dataset of protein-peptide complexes non-redundant with the AF2 training structural dataset

An initial list of protein-peptide complexes was retrieved from the PDB server[52] on April 1, 2022 with the following request: 1) Release date after May 1st, 2018 to exclude complexes present in the AlphaFold2 v.2.2 training set; 2) The longest protein (called the receptor) must contain at least 60 amino acids and the smallest chain (called the peptide) must contain at most 40 amino acids. 3) The 'Number of Polymer Instances (Chains) per Assembly' has to be between 2 and 4 and should contain heteromeric assemblies. 4) The assemblies should not contain RNA or DNA chains. The initial request led to 2484 potential candidates. Using a sequence identity threshold of 30%, we discarded all candidates for which a homolog of the receptor protein was released before May 1st, 2018 and bound to a ligand partner in the same region. From the list of selected candidates, an additional filter was used to check the absence of redundant assembly modes. For each of the selected complexes, the receptor sequence was used as a query of the PPI3D server[57], in single sequence mode, to recover all the PDB codes of complexes involving homologs of the receptor (date of PPI3D query August 2022, on the PDB updated July 20, 2022). In PPI3D, distant receptor homologs were retrieved using PSI-BLAST[58] with 2 iterations and an E-value cutoff of 0.002. For every candidate complex, PPI3D provided a detailed list of PDB codes with the chain ids involving the receptor or its homolog. We used the full list of interactions provided by PPI3D, except when it exceeded 2500 interfaces in which case the clustered list was chosen (95% sequence similarity and 50% similarity for residues in the binding region). Only the interfaces annotated as 'hetero' or 'hetero-peptide' released before May 1, 2018 were considered as potentially redundant. Their structures were compared to the candidate complex using the MM-align program (Version 20191021)[59] (option "-a") and the maximum of the three TM-scores calculated was considered. Receptor-peptide candidates with a TM-score greater than 0.5 with any other potentially redundant interface extracted from the PPI3D results were considered redundant with a previously known structure and were discarded. This latter condition only applied to structures for which MM-align successfully aligned at least 5 consecutive amino acids on the ligand side (detected by ':' in the output pair alignment corresponding to residue distance pairs <5.0 Angstrom), otherwise the interface was not considered redundant. In the end, we retained a set of 42 receptor-peptide cases to form the reference database.

### Generation of the alignments for the 42 database cases

Sequences of all the chains in the dataset of 42 complexes were retrieved from the UniProt database[60] and were submitted to three iterations of MMseqs2[61] against the uniref30_2103 database[47]. The resulting alignments were filtered using hhfilter[62] using parameters ('id'=100, 'qid'=25, 'cov'=50) and the taxonomy assigned to every sequence keeping only one sequence per species. Full-length sequences in the alignments were then retrieved and the sequences were realigned using MAFFT[63] with the default FFT-NS-2 protocol to generate the multiple sequence alignments (MSA) of every individual subunit. These MSAs, generated with full-length sequences, were then trimmed to match the delineations of the receptor and ligand parts,

which vary according to the protocols used. The sequence boundaries defined in the PDB SEQRES parameter were used to delineate the receptor and peptide binding regions. To generate the extended 100 or 200 amino acid ligands, the peptide sequence was extended in both directions, unless a chain end was encountered, in which case extension was continued in one direction only. From the individual MSAs of receptors and ligands, different types of co-alignments were generated and assessed. First, the so-called mixed co-alignments, standing for paired+unpaired co-alignments, was built by concatenating the receptor and ligand MSAs so that homologous receptor and ligand sequences were paired when they belonged to the same species (joined as if they were part of the same sequence in the alignment) and left unpaired if no common species was found (adding gaps in place of the missing homolog)[47]. Unpaired co-alignments were obtained by unpairing the paired part of the mixed co-alignments. Last, co-alignments with no evolutionary information in the ligand part were obtained from the mixed co-alignment by leaving the ligand region as a single sequence and adding gaps in the rest of the ligand alignment. In case the receptor was a heteromer or assembled as a homodimer, the multimeric assembly complex was modeled by concatenating the alignments of the receptor subunits in the same way as described above. The different concatenated co-alignments generated using the different delimitations and pairing protocols were used as input MSA to AlphaFold2.

### Generation of the input data for the scanning of the 42 ligand partners with overlapping fragments

The sequences of the 35 ligands in the non-redundant dataset that were longer than 100 amino acids, were fragmented into 100 amino acid long segments with an overlap of 30 amino acids to ensure that the binding region was entirely contained in at least one fragment. The delimitations of the tested fragments are reported in Supplementary Data 2. A multiple sequence alignment associated with each ligand fragment was built following the same protocol as above and concatenated with the alignment of the delineated binding domain of the receptor protein.

### Generation of input data for cross-partners evaluation

To generate the dataset mixing receptors and their non-cognate ligands, a subset of complexes that could be clustered according to the similarity of the type and length of the secondary structure of their ligand (reported in Supplementary Table 1) was defined (Supplementary Data 1). We selected 23 complexes with a monomeric receptor and a ligand that could be clustered into one of the 7 groups distinguished in Supplementary Data 1. The MSA of each receptor was concatenated with each ligand in the same cluster without adding MSA information on the ligand side. These alignments were used as input to generate structural models by AlphaFold2 following the protocol described below.

### Generation of the input data for the 923 cases of the ELM database

A list of ELM binding motifs and of their binding domain receptor was downloaded on July 3, 2023 from http://elm.eu.org/downloads.html. 1884 entries of receptor/ligand pairs from the ELM classes 'LIG' and 'DOC' were extracted, documented by at least one PubMed ID reference and defined delimitations for receptor and ligand binding regions (Supplementary Data 3). Of these, 492 pairs were not associated with a reference PDB structure and 469 pairs corresponded to a ligand containing multiple ELM binding motifs of the same type. The remaining 923 pairs could be used to evaluate the success rate of AlphaFold2 with protocols combining different delimitations and MSA sampling. Sequences and multiple sequence alignments of all pairs were obtained using the same procedure as described above for the 42 test cases. To define the boundaries of the binding domain in the receptor,

we could not rely on ELM delimitations as they did not always match the structural boundaries of the globular domain. Instead, we took advantage of the recently developed Chainsaw method[64] (commit tag 1ec2be5 from Jul 20, 2023) to automatically parse and assign domain boundaries from predicted receptor structures in the AlphaFold Protein Structure Database[65] (see Supplementary Data 3). The delimitations of the ELM motif in the ligand were taken from the ELM database except for motifs with less than 5 residues which were extended up to 5 amino acids (see Supplementary Data 3). Five different concatenated MSAs were generated for all receptor/ligand pairs, always using a delimited receptor domain. The ligand side was either delimited with 3 possible MSA modes (mixed, unpaired or single sequence), extended by 100 amino acids with a mixed MSA mode, or full-length ligand with a mixed MSA mode. Concatenated MSAs were used as input of AlphaFold2 to generate 25 models following the protocol described below.

### Generation of the structural models

The concatenated MSAs were used as input to run 5 independent runs of the AlphaFold2 algorithm with 3 recycles each[15,29] generating 5 structural models (25 models in total) using a local version of ColabFold v.1.3[47] with the Multimer v.2.2 model parameters[29] on NVidia V100 and A100 GPUs since the training of the v.2.2 parameters excludes complexes released after May 1st, 2018. In the case of the ELM dataset, for which the potential bias effect is present anyway, the structural models were generated using AF2-Multimer version v.2.3 and ColabFold v.1.5 with the benefit of saving time. Four scores were provided by AlphaFold2 to rate the quality of the models, the pLDDT, the pTMscore[15], the ipTMscore and the model confidence score (weighted combination of pTM and ipTM scores with a 20:80 ratio)[29]. The scores obtained for all the generated models are reported in Supplementary Data 2. The sampling of 100 models instead of 25 with a single protocol, whose performance is shown in Supplementary Fig. 4b, was achieved by increasing the number of independent runs from 5 to 20. No relaxation step was performed consistently with our own approach and since relaxation has been found to be computationally costly with little added value to the quality of results[15]. To evaluate the models generated from the sampling of fragments of 100 amino acids along the full ligand sequences, we selected the model with the highest ipTM score (see Supplementary Data 2), in order to focus on the interface and be less dependent on the degree of folding in the fragment itself, reflected in the pTM score. To evaluate the models from the ELM dataset, we selected the model with the highest AF2 confidence score among the 25 models generated with each of the tested protocols (scores reported in Supplementary Data 3).

### Testing of the AFsample protocol

AFsample was cloned from https://github.com/bjornwallner/alphafoldv2.2.0 (commit tag 9f76c2a from Dec 24, 2022). AFsample was run on a selection of cases (see below) following the procedure outlined in the provided pipeline script (run_afsample.sh). (i) Create the MSAs and search for templates (hence the MSAs used are different in the AFsample runs and in our approach). (ii) Generate 6000 models (or as many as was possible given the computational cost) sampled following four schemes (a) 10 ×200 models using multimer_v1 & multimer_v2 parameters, using dropout and templates, (b) 10 ×200 models using multimer_v1 & multimer_v2 parameters, using dropout and no templates, (c) up to 5 ×200 models using multimer_v1 parameters, using dropout, no templates, 21 recycles (denoted r21), and (d) up to 5 ×200 models using multimer_v2 parameters, using dropout, no templates, 9 recycles (denoted r9). (iii) Sort all models according to their combined score. Since AFsample is very computationally intensive, we targeted two lists of cases for testing, with the goal to answer two questions about the complementarity between AFsample and our approach: (i) 17 cases with combined score lower than 0.8 for the rank 1 model in our mixed-delim-delim approach (independently of whether

this approach succeeds or fails). This list addresses the question of the relative success rates for AFsample and our approach, following the recommendation of the AFsample publication that their pipeline should be run only when the best ranked solution has combined score lower than 0.8. (ii) 10 cases failing with our mixed-delim-fl pipeline and with total size (cumulated over the complex partners) smaller than 1000 amino acids (above 1000, the AFsample runtime becomes prohibitive, see Supplementary Data 6). This list addresses the question whether the exhaustive AFsample approach can succeed where our approach fails.

### Evaluation and visualization of the structural models

The structural models generated with every alignment protocols were compared to their reference structure defined in Supplementary Table 1 for the 42 cases of the non-redundant dataset and in Supplementary Data 3 for the ELM dataset. The structural models were first cut following the delimitations of the reference experimental structure to ensure proper superposition of receptors and ligands models. For the evaluation of the ELM models dataset, the reference structure was, if available, the PDB structure exactly matching the sequence of the receptor/ligand couple. In case such an experimental structure was not available, we tested all possible reference PDB structures belonging to the same ELM TYPE category and evaluated the accuracy of the models using the reference PDB with the highest DockQ score (listed in Supplementary Data 6). In case there was internal symmetry in the receptor bound asymmetrically by a ligand (as in the case of coiled coils), we used different reference structures for each symmetrical arrangement (which we provide in the Zenodo archive) and selected the one that provided the best DockQ score.

The accuracy of all models was assessed using two related measures (i) the DockQ score, which provides a continuous value between 0 and 1, with limits of 0.23, 0.49, and 0.8 defining Acceptable, Medium, and High quality thresholds for protein-protein complexes[48] (ii) the more stringent conditions established by the CAPRI community to rate the specific cases of receptor-peptide complexes using ligand and interface Root-Mean-Square Deviation (L- and iRMSD) and the Fraction of native contacts (fnat). Ranks are assigned depending on the following criteria: High (fnat in [0.8, 1.0] and (L-RMSD ≤ 1.0 Å or iRMSD ≤ 0.5 Å)), Medium (fnat in [0.5, 0.8] and (L-RMSD ≤ 2.0 Å or iRMSD ≤ 1.0 Å) or fnat in [0.8, 1.0] and (L-RMSD > 1.0 Å and iRMSD > 0.5 Å)) and Acceptable (fnat in [0.2, 0.5] and (L-RMSD ≤ 4.0 Å or iRMSD ≤ 2.0 Å) or fnat [0.5, 1.0] and (L-RMSD > 2.0 Å AND iRMSD > 1.0 Å))[49]. Additional analyses were performed following the standard metrics calculated by CAPRI assessors to rate the similarity between the models and their reference structure (such as the fraction of interface residues FRIR or the fraction of non-native contacts FRNNAT) and are also available in Supplementary Data 1 and Supplementary Data 3 for the 42 non-redundant and the ELM dataset, respectively. 3D structures were visualized and represented using ChimeraX[66].

### Reporting summary

Further information on research design is available in the Nature Portfolio Reporting Summary linked to this article.

## Data availability

Source data used to generate all Figures and Supplementary Figures are provided as a Source Data file. The reference PDB files of the 42 test cases, the multiple sequence alignments built for all ten protocols and the corresponding PDB files of the predicted models have been deposited[67] in the ZENODO database under the accession DOI code https://doi.org/10.5281/zenodo.7838023 [https://zenodo.org/doi/10.5281/zenodo.7838023]. The reference PDB files used to evaluate the predictions of the 923 cases from the ELM database, the multiple sequence alignments built for all five protocols and the PDB files of the

models predicted with highest scores have been deposited[67] in the ZENODO database under the accession DOI code https://doi.org/10.5281/zenodo.7838023 [https://zenodo.org/doi/10.5281/zenodo.7838023]. Source data are provided with this paper.

## Code availability

The code for processing, analyzing and visualizing the results is available at: https://github.com/i2bc/SCAN_IDR and the version used was also deposited[68] in the ZENODO database under the accession DOI code https://doi.org/10.5281/zenodo.10213747 [https://zenodo.org/doi/10.5281/zenodo.10213747].

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

## Acknowledgements
We thank Sjoerd de Vries and Isabelle Callebaut for useful discussions. The work was supported by grants from Agence Nationale de la Recherche (ANR-21-CE44-0009-01 to R.G. and ANR-18-CE45-0005-01 ESPRINet to J.A.). This work was granted access to the HPC resources of IDRIS under the allocation 2023-AD010314343 to R.G. made by GENCI and to the BIOI2 platform resources at the I2BC.

## Author contributions
H.B., J.A. and R.G. designed the study. H.B. built the dataset and developed the scripts to generate the alignments, ran AlphaFold2 and analyzed the outputs. J.G. and D.J.Z. contributed, respectively, to the scanning protocol and to the analysis of the ELM database. J.A. and R.G. contributed to the script development and data analysis, jointly supervised the work and wrote the manuscript together with H.B.

## Competing interests
The authors declare no competing interests.
