## [Peer Review File · Nature Communications]

From interaction networks to interfaces, Scanning intrinsically disordered regions using AlphaFold2REVIEWER COMMENTS

Reviewer #1 (Remarks to the Author):

Bret et al. report studies based on AlphaFold2-Multimer that aim to understand the predictive capabilities of this method for protein-peptide complexes involving disordered regions. They curate a new non-redundant database of 42 complexes deposited in PDB after 2018 to avoid using structures used in AlphaFold2 training. One of the primary outcomes of this work is to demonstrate the limited success when full-length protein sequences are used and how one can overcome this by using smaller fragments. Furthermore, the authors present the results from different strategies for using MSA alignment and how the results can differ. Using different alignment modes in conjunction is also shown to improve the success in predicting the correct protein-peptide binding. Lastly, specific and non-specific peptides/ligands are used to test the ability to predict specificity.

Overall, the paper is timely as the use of AlphaFold2-Multimer is increasing rapidly. It is also well-written and accessible to a general audience, except for the part about mixed co-alignments and incorporating the evolutionary information, which could use more information.

I have the following comments for the authors to consider as they revise the manuscript.

1. It appears that the authors focus on the known binding region buried inside an extended sequence when testing the model initially, and the success rates reported are based on some implicit assumptions. Even though later in the paper, the authors do cross-testing, why not just feed overlapping or non-overlapping fragments of the ligand peptide and see if it finds the binding region? Doing this for at least a handful of cases may be sufficient to know how well the model performs.
2. While discussing the misleading assignment on page 17, the authors seem to ignore that these so-called non-specific peptides may bind the receptors unless I am missing some information about this in the methods. Are there any biophysical experiments that show the lack of binding?
3. At several places in the paper where the authors discuss the potential reasons for not getting a successful prediction, I would have liked to see some discussion about the possible paths forward. Are these issues insurmountable and intractable with AI-based methods?

Reviewer #2 (Remarks to the Author):

In this paper, the authors show that AlphaFold2.2 can identify disordered binding regions by scanning these against a potential binder. This is, in principle, an extension to the earlier work on peptide-protein docking using AlphaFold, but using not only the peptide bound but entire regions.

The authors have used a rigorous method to identify the overlap between the training set of AlphaFold and their test set. This is excellent. Unfortunately, this also reduced the size of the test set significantly to only 42 pairs.

Major:

The authors point out that alternative methods to pair MSAs might help. This is somewhat similar to what Waller (Ref #40) does in his method (AFsample), which was the best method in CASP15 for docking. A comparison with this would be valid.

For this method to be truly useful, it should be demonstrated that it can be used to identify novel binders. The ELM resource (ref #4) is a good starting point. Using the 359 known binding domains (and corresponding peptides) would be computationally possible to see how many of these could be identified.

Minor:

In supply Fig 3 it would be much easier to compare if the PDB codes were sorted equally in the three panels.

It is not entirely clear how DockQ is used when the full-length ligand/receptors are used. By default, DockQ uses needle to align the sequences of both receptor and ligand, but this sometimes fails to identify the correct residues (repeated sequences, long gaps in the PDB files etc). Has the authors checked that the correct regions are used?

How many of the complexes (for different delimitations) were correct if you took the best out of all 100 models generated with different MSAs?

Point-by-point response to the reviewers' comments

We would like to thank the reviewers for their insightful comments and requests. We have carefully addressed all the points that were raised. Briefly, we integrated novel data presented and discussed in the main manuscript and in a number of additional Supplementary Materials along four major axes:

- Demonstration that the fragment scanning strategy using overlapping fragments of 100 amino acids is successful in detecting the correct location of the binding motif.
- Expansion of the observed results from 42 test cases to a much larger dataset of 923 receptor/ligand pairs taken from the ELM dataset.
- Comparison with the AFsample method.
- Impact of increasing the sampling of every protocol from 25 to 100 models.

In addition, during revisions, we found that for one of the 42 non-redundant cases, we had selected the wrong isoform compared to the one used in the original PDB (6J08). The isoform whose structure was resolved was Q3KP22-3, whereas we used the Q3KP22-1 isoform by default, since isoform indexes are not indicated in the PDB database. From a case in which the receptor could not be folded by AF2, this case with the correct isoform turned into a successful prediction in most of the protocols we tested and consequently increased the success rates we had previously reported. We have corrected all previous results tables and figures accordingly.

All parts of the main manuscript that have been modified from the original version are shown in red to facilitate the second review. We hope that this revised version will provide the information that was missing in the initial manuscript.

- **Reviewer #1 (Remarks to the Author):**

Bret et al. report studies based on AlphaFold2-Multimer that aim to understand the predictive capabilities of this method for protein-peptide complexes involving disordered regions. They curate a new non-redundant database of 42 complexes deposited in PDB after 2018 to avoid using structures used in AlphaFold2 training. One of the primary outcomes of this work is to demonstrate the limited success when full-length protein sequences are used and how one can overcome this by using smaller fragments. Furthermore, the authors present the results from different strategies for using MSA alignment and how the results can differ. Using different alignment modes in conjunction is also shown to improve the success in predicting the correct protein-peptide binding. Lastly, specific and non-specific peptides/ligands are used to test the ability to predict specificity.

Overall, the paper is timely as the use of AlphaFold2-Multimer is increasing rapidly. It is also well-written and accessible to a general audience, except for the part about mixed co-alignments and incorporating the evolutionary information, which could use more information.

>>> Thank you for your positive report. We tried to reformulate how we incorporate evolutionary information into the mixed co-alignments in Methods with the hope it would make the explanations more accessible. We added a Supp. Figure 2b to illustrate this and added reference to the Method section in Results where this issue is raised and added references to the Supp. Figure 2a and 2b in the Methods . We hope it addresses better the concern of accessibility.

I have the following comments for the authors to consider as they revise the manuscript.

1. It appears that the authors focus on the known binding region buried inside an extended sequence when testing the model initially, and the success rates reported are based on some implicit assumptions. Even though later in the paper, the authors do cross-testing, why not just feed

overlapping or non-overlapping fragments of the ligand peptide and see if it finds the binding region? Doing this for at least a handful of cases may be sufficient to know how well the model performs.

>>> We would like to thank reviewer 1 for this excellent suggestion. This test was indeed lacking prior to cross-testing between multiple binders, as it challenges the performance of the approach to scan a pair of known interactors. We set up a systematic scanning analysis of the 42 full-length ligands of the non-redundant dataset and assess the efficiency of recognizing the correct fragment relative to other regions of the ligand protein. Fragments of 100 amino acids overlapping by 30 amino acids were considered as a fair trade-off between sensitivity and the number of runs to be performed. For 7 of the 42 complexes, the ligand size was less than 100 aa and they were disregarded. For the 35 remaining cases with length over 100, the results show that in 31 cases (89 % success rate), the fragment with the highest ipTM score contained the correct binding region. We have added Figure 4 and Supp Table 3 to present and discuss these results (**p.12, l.16 – p.14, l.11**). In particular, of the 4 missed cases, 39_7O6N failed because we did not model the receptor as a homodimer (Supp. Figure 5), the incorrect 100 amino acid fragments tend to bind to the accessible interface of the homodimer, generating incorrect interfaces with a significant confidence score. This underlines the importance of representing the biological context as closely as possible, taking into account the assemblies of homomers and heteromers pre-existing the formation of an interaction with a disordered region as highlighted in the discussion (**p. 25, l. 9 – 13**). Overall, this additional analysis shows that analysis of a pair of interactors using the fragment scanning strategy is highly valuable, as it is unlikely that a wrong competitive binding site will be found in a ligand protein.

2. While discussing the misleading assignment on page 17, the authors seem to ignore that these so-called non-specific peptides may bind the receptors unless I am missing some information about this in the methods. Are there any biophysical experiments that show the lack of binding?

>>> Unfortunately, no biophysical experiment has demonstrated the absence of binding. We have added a note in the manuscript to this effect (**p. 21, l. 4 – 8**):

“For a few specific classes of binding motifs, a recent comparison with experimental data also indicated a lack of specificity for AF2 predictions⁵¹. However, in the absence of further biophysical experiments, we cannot completely rule out that, for the misleading assignments discussed above, non-specific peptides may indeed exhibit detectable binding to their non-cognate receptors.”

Residual binding may generally exist, particularly in the case of amphiphilic helices binding to a receptor. Analysing the potential ‘stickiness’ of a receptor of interest by testing different independent peptides could help to gain more confidence for that receptor. Other strategies are discussed following this reviewer's next comment.

3. At several places in the paper where the authors discuss the potential reasons for not getting a successful prediction, I would have liked to see some discussion about the possible paths forward. Are these issues insurmountable and intractable with AI-based methods?

>>> We have added a section to the discussion on ways to improve correct binding mode discriminations.

- First, paying attention to the potential homomultimeric or heteromultimeric properties of certain receptors can greatly aid in generating correct binding modes for peptide ligands (discussed **p. 25, l. 9 – 13**). We carefully inspected the 42 non-redundant test cases in this respect and this improved performance.

- Regarding the lack of specificity, we also discuss three different tracks to improve the confidence of a prediction (discussed **p. 27, l. 7 – 15**) i) the use of competition simulations as proposed by Chang and Perez (ref. 55) ii) the use of AF2rank implemented as a colab by Roney and Ovchinnikov (ref. 54) which has recently been adapted to multimer scoring (using a model structure of the complex and a single pass of AF2 without iteration), iii) the assessment of receptor 'stickiness' by testing several unrelated peptides with similar secondary structures. We are aware that another potential reason limiting the sensitivity of the predictions is the risk of the binding sites being masked by internal regions of the receptor protein, in the same way as we saw masking on the ligand side in Figure 6bcd. However, we did not observe any such case in our dataset, hence we did not discuss this in detail in the manuscript text.

“Using the strategy described in ⁵⁵ could be a way to circumvent some specificity issues. Alternatively, rescoring complex models for different peptides using the updated AF2rank colab may provide complementary discriminative power ⁵⁴. Some receptors may also show more promiscuous binding properties than others when assessed from AF2 confidence score as shown in the case of 7CZM. Using a set of representative peptides such as those used in this study, it may be possible to spot out receptors more prone to interacting non-specifically with various motifs and improve normalization of the confidence score.”

- Additional possible paths forward to further improve the success rate of AF2 for complexes involving IDRs include increased sampling, as discussed **p. 26, l. 4 – 16** following Reviewer #2's suggestion to compare our approach with AFsample.

- **Reviewer #2 (Remarks to the Author):**

In this paper, the authors show that AlphaFold2.2 can identify disordered binding regions by scanning these against a potential binder. This is, in principle, an extension to the earlier work on peptide-protein docking using AlphaFold, but using not only the peptide bound but entire regions.

The authors have used a rigorous method to identify the overlap between the training set of AlphaFold and their test set. This is excellent. Unfortunately, this also reduced the size of the test set significantly to only 42 pairs.

>>> Thank you very much for this positive feedback and for raising the concern about the test set size that we met because we wanted to ensure that our conclusions could not be biased by the presence of training samples. Following reviewer 2 recommendation, we believe we are now providing a more thorough assessment of considering shorter fragments when scanning intrinsically disordered regions, by complementing our unbiased dataset of 42 pairs with a larger-scale analysis of ELM motifs with the risk that some of the predicted cases might be biased by their similarity to cases used in the training of AF2-Multimer parameters.

Major:

The authors point out that alternative methods to pair MSAs might help. This is somewhat similar to what Waller (Ref #40) does in his method (AFsample), which was the best method in CASP15 for docking. A comparison with this would be valid.

>>> We have performed a thorough comparison with the AFsample protocol published in Bioinformatics last September by B. Wallner. Given the massive amount of computer power required to run this approach, involving the generation of 6000 models (rather than 100 in our combined protocol), it was not possible to run all the targets of the 42 non-redundant dataset. Rather, we considered two subsets of 17 and 10 targets addressing two specific questions detailed below. In the manuscript, we added a detailed description of the performance in Supp Table 8a and 8b and added a section to the discussion showing the interest of the comparison (discussed **p. 26, l. 4 – 16**):

“To further enhance the chance of generating a correct solution using short delimited peptides as ligands, we also explored how the AFsample strategy would perform on some of the difficult cases found among the 42 cases in our non-redundant dataset (see Methods). Our results highlight a complementarity of the two approaches, where some cases were successful with our combined protocol but not with AFsample, while others were unsuccessful in our combined protocol and solved by AFsample, albeit at a much larger computational cost (Supp Table 8a). A comparative analysis of the two approaches suggests a few guidelines that could be used to further increase success rates: the use of templates, of a combination of multimer_v1 and multimer_v2 parameters and a larger sampling for a given condition (up to 200 models per condition instead of 25). In contrast, on our dataset, using a larger number of recycles as in AFsample was never necessary to obtain successful predictions (using 9 or 21 recycles did not improve success rates). Additionally, if we had stopped sampling after 200 models for each condition instead of 1000 as implemented by default in AFsample, we would have obtained the same best models.”

We first analyzed a subset of 17 cases because they had an AF2 confidence score lower than 0.8 for the rank 1 model in the SCAN_IDR mixed-delim-delim approach (independently of whether this approach succeeds or fails). This subset addressed the question of the relative success rates for AFsample and our SCAN_IDR approach on small well-delimited ligands. In two cases, AFsample managed to identify a correct solution that was not sampled with our combined approach. In two cases, AFsample did not reach any correct solution while they were successfully predicted by our combined strategy. Interestingly, for the two cases that we missed, AFsample succeeded because it used either templates or multimer_v1 parameters, options that can be easily added to the options used in the fragment scanning combined approach to further increase the sensitivity of the detection.

The second subset of 10 cases that was analyzed with AFsample, focused on the possibility to use AFsample instead of the ligand scanning protocol that we implemented in Figure 4. These cases were selected because they failed with our ‘mixed-delim-fl’ condition (delimited receptor and full-length ligand using mixed alignment) and contained less than 1000 amino acids. Indeed, AFsample can require prohibitive computer resources required when applied to larger systems (a single case of size 977 already required 300 GPU hrs, Supp Table 8b). AFsample found a correct solution for only 3 out of ten cases and again the use of multimer_v1 was found instrumental.

Therefore, given the high computational cost for running AFsample for largely disordered regions, it appears highly reasonable to apply instead our fragment scanning approach and combined protocol using different MSA inputs and delimitations to reduce the complexity of the search.

For this method to be truly useful, it should be demonstrated that it can be used to identify novel binders. The ELM resource (ref #4) is a good starting point. Using the 359 known binding domains (and corresponding peptides) would be computationally possible to see how many of these could be identified.

>>> Thanks for this sound suggestion to expand the scope of our study. We carefully considered the proposition of reviewer 2 by a detailed analysis of the ELM database. We specifically considered the

LIG or DOC classes in the ELM database enriched in generic small linear binding motifs. Among the LIG and DOC categories, we selected those for which the interaction was validated with a pubmed ID and the definition of precise delimitations (1884 out of 2485 receptor/ligand pairs). Among these pairs, the subset possessing (i) a unique binding site in the ligand and (ii) a PDB reference for evaluation (either exact or homologous) contained 923 cases divided into 84 categories of ELM types, corresponding to different families of domains and their associated consensus motifs (Figure 8a). We ran the predictions for this entire set of protein pairs (all information and results are provided in Supp. Table 5), testing 5 protocols where the receptor domain is delimited and the ligand delimitations can be either (i) full-length (ii) delimited binding site extended by 100-aa (iii) delimited binding site with 3 MSA modes (mixed, unpaired and with no alignment). We also evaluated the interest of the combined protocol as carried out for the 42 non-redundant test cases. One of the difficulties in handling the ELM database is that the delimitations are not always properly matching the delimitations of the globular receptor domain and can miss some secondary structure elements. Incomplete structures of receptors are highly detrimental to the success of AF2 predictions since the ligand peptides tend to replace the missing secondary structure elements. Fortunately, we could rely on the chainsaw method (released as we were processing the revisions) to automatically delimitate the receptor domains with high confidence. All the delimitations used for the different conditions together with the original information provided in the ELM database are listed in Supp. Table 5.

For the global evaluation, we focused on the cases where a single binding site existed in the ligand. Indeed, it was difficult to fairly compare the full-length condition (where only one peptide can bind) with the other categories (allowing that all the binding sites be detected). This corresponded to 84 different ELM types, 196 cases with an exact PDB structure and 727 with a homologous template which could be used to rate the accuracy of the prediction. Overall success rates were similar to those obtained for the 42 cases (Figure 8b and Supp. Figure 8). The lack of sensitivity of AF2 in the case of full-length ligands and the favourable effect of combining the 4 alignment protocols could be observed on a much larger scale. Interestingly, cases with exact PDB structures tend to be much better predicted, especially in conditions where the ligand is considered full-length, which may potentially be due to their possible presence in the training dataset of AlphaFold.

Minor:

In supply () Fig 3 it would be much easier to compare if the PDB codes were sorted equally in the three panels.

>>> To correct this inconvenience, we kept the same order based on the size of the full-length systems (see revised Supp. Figure 3).

It is not entirely clear how DockQ is used when the full-length ligand/receptors are used. By default, DockQ uses needle to align the sequences of both receptor and ligand, but this sometimes fails to identify the correct residues (repeated sequences, long gaps in the PDB files etc). Has the authors checked that the correct regions are used?

>>> This is a point that we indeed carefully examined when comparing the models with the reference structures. Once generated under full-length or other extended conditions, the models were systematically trimmed to ensure that the amino acids present in the models exactly matched those provided in the reference PDB. The trimmed models were renumbered to check that the alignment with the reference PDB was correct. Beyond the potential pitfall mentioned by the reviewer, one of the problems encountered was also to find the correct way to superimpose the solutions when there

was internal symmetry in the receptor bound asymmetrically by a ligand (as in the case of coiled coils). In the latter case, we used different reference structures (which we provide in the zenodo archive) and selected the one that provided the best DockQ score. All these evaluation protocols have been developed and calibrated over the years of our participation in the CAPRI challenge and we have developed this part with an awareness of the many pitfalls that could indeed arise from the evaluation of models of complexes involving bound peptides.

How many of the complexes (for different delimitations) were correct if you took the best out of all 100 models generated with different MSAs?

>>> We're not sure we interpreted this question correctly. We understand that the reviewer was concerned that the comparison between the combined approach (based on 100 models) and the individual MSA methods (based on 25 models) might be biased by the different number of models taken into account. First of all, we would like to point out that the combined approach was designed by taking exactly the same models as those obtained with the 25 models of the 3 MSA modes with delimited peptide (mixed, unpaired, single sequence) as well as the 25 models obtained with the mixed MSA extended to 100. To control for potential sample size bias, we repeated the predictions by generating 100 models with each of the 4 individual MSA protocols, and present the results in Supp. Figure 4b. We found that increasing the number of models without changing the MSA construction modes and delimitations did not change the success rates, in contrast to the significant increase obtained by combining 4 different protocols each generating 25 models to end up with 100 models.

REVIEWERS' COMMENTS

Reviewer #1 (Remarks to the Author):

The authors have done a wonderful job addressing previous comments from the reviewers. In fact, they have gone beyond what I was expecting and invested a lot of effort to carefully assess the issues raised.

Reviewer #2 (Remarks to the Author):

The authors have satisfactorily answered my concerns

REVIEWERS' COMMENTS ON THE REVISED MANUSCRIPT

Reviewer #1 (Remarks to the Author):

The authors have done a wonderful job addressing previous comments from the reviewers. In fact, they have gone beyond what I was expecting and invested a lot of effort to carefully assess the issues raised.

Reviewer #2 (Remarks to the Author):

The authors have satisfactorily answered my concerns

AUTHORS' COMMENTS TO THE REVIEWERS' COMMENTS

We are very grateful to the reviewers for their careful review of our manuscript and for the excellent suggestions made during the revision process.